# Contamination of Foods from Cameroon with Residues of 20 Halogenated Pesticides, and Health Risk of Adult Human Dietary Exposure

**DOI:** 10.3390/ijerph18095043

**Published:** 2021-05-10

**Authors:** Yamdeu Joseph Hubert Galani, Michael Houbraken, Abukari Wumbei, Joseph Fovo Djeugap, Daniel Fotio, Yun Yun Gong, Pieter Spanoghe

**Affiliations:** 1School of Food Science and Nutrition, University of Leeds, Leeds LS2 9JT, UK; y.gong@leeds.ac.uk; 2Department of Plants and Crops, Faculty of Bioscience Engineering, Ghent University, Coupure Links 653, 9000 Ghent, Belgium; michael.houbraken@aphea.bio (M.H.); awumbe2003@yahoo.com (A.W.); Pieter.Spanoghe@ugent.be (P.S.); 3Department of Plant Protection, Faculty of Agronomy and Agricultural Sciences, University of Dschang, Dschang P.O. Box 222, Cameroon; joseph.djeugap@univ-dschang.org; 4Inter-States Pesticides Committee of Central Africa, Yaounde P.O. Box 16344, Cameroon; danfotio@yahoo.co.uk

**Keywords:** food safety, POPs, organochlorine, pesticide residues, risk assessment, Cameroon

## Abstract

(1) Background: Halogenated pesticides are abundantly used in Cameroon, but there is no information on the health risk of consumers from exposure to their residues in foods. (2) Methods: Residues of 20 halogenated pesticides were determined in 11 agricultural products collected in the 3 largest cities of Cameroon using QuEChERS extraction and gas chromatography with electron capture detector (GC-ECD), and health risk from dietary exposure was assessed. (3) Results: Organochlorines pesticides aldrin, p,p’-dichlorodiphenyl-trichloroethane (DDT) and β-hexachlorocyclohexane (β-HCH) found in 85.0%, 81.9% and 72.5% of samples, respectively, were the most frequently detected. The highest average concentrations of residues were 1.12, 0.74 and 0.39 mg/kg for methoxychlor, alachlor and β-HCH, respectively, found in chilli pepper. Chili pepper (58.9%), cowpea (56.8%), black beans (56.5%) and kidney beans (54.0%) exhibited the highest residue occurrences. Levels above the European Union maximum residue limits (MRLs) were found for all the 20 pesticides, in 40.1% of the positive analyses, and the food samples contained 14 pesticides banned in Cameroon. Chronic, acute, cumulative and carcinogenic risk assessments revealed that lifetime consumption of maize, black beans, kidney beans, groundnuts and chili pepper contaminated with aldrin, dieldrin, endrin, HCB, heptachlor, o,p’-DDT, p,p’-DDD, p,p’-DDT, p,p’-DDE and β-HCH, could pose health risks. (4) Conclusion: These results show that there is an urgent need of pesticide usage regulation, effective application of pesticide bans and management of obsolete pesticide stocks in Cameroon.

## 1. Introduction 

In the development of modern and innovative pesticides in agricultural chemistry during the past 35 years, halogen-containing active ingredients have had a significant role: around 96% of the herbicides, fungicides, insecticides/acaricides and nematicides products launched between 2010 and 2017 contained halogen atoms [1]. Organochlorine pesticides (OCPs) are the most used halogen-containing agrochemicals worldwide, and like most of the other halogenated pesticides, they present significant environment and health problems [2,3,4]. Exposure to OCPs poses severe health threats to biota, including humans [5,6,7]. Some OCPs such as heptachlor and hexachlorobenzene (HCB) are classified Group 2B (possibly carcinogenic to humans) while others, including dichlorodiphenyl-trichloroethane (DDT), hexachlorocyclohexane (HCH), aldrin and dieldrin are Group 2A (probably carcinogenic to humans) [8]. For these reasons and added to their persistence and pollution of the environment, restriction and elimination of harmful persistent organic pollutants (POPs), including OCPs, were acted at the Stockholm Convention on POPs in 2001 [9]. Nevertheless, in developing countries, the use of many other OCPs has been rising [4]. Human exposure to pesticides may occur via several routes, including dermal absorption and air inhalation [10,11], but the ingestion of contaminated food accounts for more than 90% of total exposure [12]. Therefore, aside of the measures for controlling their environmental impact, it is important to monitor these pesticides in foods and assess the health risk they may pose to consumers. 

Survey studies have shown that pesticides, including halogenated compounds, are intensively used in Cameroon by farmers and traders to protect their plants and products during production and postharvest storage [13,14,15,16,17,18]. Organochlorines such as DDT have been extensively used in Cameroon for both malaria vector control and agricultural purposes in the 1950s through to the 1960s before being replaced by pyrethroids like cypermethrin and deltamethrin [19,20]. A number of factors that can contribute to high exposure of consumers such as intensive utilization by farmers and limited knowledge about pesticide use are reported in Cameroon [14,16,17,18,21,22,23,24,25]. There is also a high number and different types of obsolete pesticides accumulated over the years in the country, among which are many halogenated compounds [26,27]. Further, with little or no control around these pesticide stocks, they could be a source of severe acute or chronic pollution and toxicity [28] or can be used fraudulently by farmers [17,29]. For instance, banned halogenated compounds including DDT and its metabolites were found in food samples from Cameroon [23,30]. Although the illegal application is suspected [29], the origin of DDT found in food samples in Cameroon remains unknown. 

Despite this increased concern on the risk of dietary exposure and health problems linked to halogenated pesticides in Cameroon, little has been done for assessing how these pesticides can impact food safety and consumers health in the country, probably because of lack of food consumption data. The last study in Cameroon was performed 12 years ago and reported that dietary exposure to pesticide residues in the capital city Yaoundé was low. But the authors recommended that further investigations should be done using more sensitive analytical methods [28]. More recent studies found pesticide residues above the maximum residue limits (MRLs) in maize, cowpea and millet samples from northern Cameroon, and banned OCPs endosulfan, DDT and HCH were found in cowpea and derived products [23,29,31]. Furthermore, our previous study showed that in 12 agricultural products from western highlands of Cameroon, at least one pesticide was found in all the samples, 34.4% of the tested compounds exceeded their European Union MRLs, and halogenated compounds were the most frequently quantified pesticides and at high concentrations [30]. Meal samples from Douala and Garoua in Cameroon contained residues of organophosphate and organochlorine pesticides [32]. All these studies suggested that human dietary exposure is potentially high, and continuous monitoring and dietary risk assessment of pesticide residues in Cameroon is an urge. 

Cocoa, coffee, palm oil, maize, beans, cassava, groundnuts, plantains and bananas are the most important food commodities in Cameroon, together with other items largely produced and consumed items like soybean, chili pepper, Egusi seeds and white pepper [33]. More attention is being given to dried foods in pesticide monitoring programs because they are concentrated than fresh agricultural products and therefore may contain higher pesticide residue levels [34]. However, no report can be found on levels of halogenated pesticide residues in dried food commodities harvested in Cameroon and found in the markets of main cities of the country. In addition, since 12 years, there is little information on the health risk assessment of pesticide residues from dietary sources in the country, and no attention have been paid to halogenated pesticides. Hence, the present study determined residues of 20 halogenated pesticides in 160 samples of 11 agricultural products collected in the three largest cities (Bafoussam, Douala and Yaounde), extracted using QuEChERS method and analyzed by gas chromatography with electron capture detection (GC-ECD). The pesticide residues data were then combined with the Cameroon food consumption data from the Sub-Saharan Africa Total Diet Study (SSA-TDS) [35] to assess whether intake of these foods by Cameroonian consumers could be of health concern. 

## 2. Materials and Methods 

### 2.1. Sample Collection 

Samples of 11 dried agricultural produced, i.e., black beans, chili pepper, cocoa beans, coffee beans, cowpea, Egusi seeds, groundnuts, kidney beans, maize, soybeans and white pepper were collected in the main markets of Bafoussam, Douala and Yaounde, in December 2017. These food items were selected because they represent the main agricultural commodities produced and consumed in Cameroon. The three selected cities are the largest, most populated and most diversified cities of the country. For each food item, five samples were collected in each market, but coffee beans were not available in Yaounde because this is not a coffee-growing area in the country, therefore, 160 samples were obtained in total. Approximately 300 g of each food was taken in a hard paper envelope within polyethylene plastic bags, after mixing the whole lot available at the merchant store to obtain a representative sample. The bags were sealed, labelled, transported to the Laboratory of Crop Protection Chemistry at Ghent University, Belgium, where they were kept at 20 °C. 

### 2.2. Pesticide Residues Analysis 

The list of registered agricultural pesticides authorized in Cameroon and banned compounds in the country, established by the Ministry of Agriculture and Rural Development, and recommended to be applied on the sample crops [36] was used to select the active ingredients to be screened. The selected 20 halogenated compounds included 12 organochlorines, 4 pyrethroids, and 1 chloronitriles, phenylamides and phtalimides. There were 1 herbicide, 3 fungicides and 16 insecticides (Table 1). Pesticides extraction, dispersive solid phase extraction (d-SPE) clean-up, and quality control were performed using the QuEChERS method, and analysis was achieved by GC-ECD using the method previously developed and validated by [30]. 

#### 2.2.1. Chemicals 

For extraction and clean-up, analytical grade reagents of at least 99% purity were used. UPLC-grade acetonitrile and hexane were obtained from VWR Chemicals (Leuven, Belgium). Salts (disodium hydrogen sesquihydrate, trisodium citrate dehydrate, sodium chloride and anhydrous magnesium sulfate) and the pesticide active ingredient standards were procured from Sigma-Aldrich (Bornem, Belgium). Clean-up tubes (d-SPE) were procured from Waters (Zellik, Belgium). A Milli-Q purification system was used to produce water. 

#### 2.2.2. Extraction and Clean-Up of Pesticide Residues

Approximately 50 g sample was powdered in a household mill (Krups, Fleurus, Belgium), then 5 g of powder was weighed into a 50 mL Teflon capped centrifuge tube, 5 mL of Milli-Q water and 15 mL of acetonitrile were added, and the mixture was vigorously shaken for 1 min. A mixture of salts, including 6 g anhydrous magnesium sulfate, 1.5 g trisodium citrate dihydrate, 1.5 g sodium chloride, and 0.75 g disodium hydrogen citrate sesquihydrate was added to the extract in the tube, which was agitated for 3 min at 300 rpm on a shaker (Edmund Bühler, Hechingen, Germany). After centrifugation for 5 min at 10,000 rpm (Eppendorf, Leipzig, Germany), the supernatant was collected. Samples of groundnuts, chili pepper, coffee, cocoa, and white pepper underwent clean-up to remove contaminants that could interfere with the analysis. For clean-up, 8 mL of the supernatant was pipetted into a 15 mL d-SPE tube packed with 300 mg primary secondary amines (PSA), 900 mg MgSO4, and 150 mg octadecyl (C18), the mixture was shaken for 1 min, centrifuged for 5 min at 3000 rpm, and the supernatant was collected. For GC-ECD analysis, solvent was changed by taking 5 mL of supernatant into a 10 mL flask and acetonitrile was evaporated (Heidolph Instruments, Schwabach, Germany) at 40 °C until dryness. The solvent was replaced by 5 mL of hexane, and 2 mL of the extract was sampled into a crimp top autosampler vial for analysis.

#### 2.2.3. Residues Analysis by Gas Chromatography with Electron Capture Detection

The halogenated compounds were evaluated on a gas chromatograph (Agilent 6890N Network) with an electron capture detector (Agilent Technologies, Diegem, Belgium). Compounds were separated through a HP-5MS (5% phenyl methyl siloxane) capillary column (30 m × 0.25 mm, 0.25 μm) set at the following operating conditions: initial column temperature was 80 °C, it was increased to 205 °C at a rate of 30 °C/min, and held for 4 min, then further increased to 290 °C at a rate of 20 °C/min and held constant for 8 min, followed by another increase to 325 °C at a rate of 50 °C/min. The injector was maintained at 280 °C and the detector at 300 °C. The carrier gas was helium, used at a flow rate of 1.1 mL/min. samples were injected in the split mode with a split ratio 52.7:1. System control, data acquisition and data analysis were performed with Agilent Software GC ChemStation version Rev. A.10.02. 

#### 2.2.4. Quality Control of Residues Analysis

The analysis was validated according to European Commission recommendation SANTE/11945/2015 [37], with 8 replicates of samples obtained from organic agriculture markets in Belgium, spiked with pesticide standards at 0.01 mg/kg. The spiked samples were allowed 1 h for pesticide absorption into samples, then they were subjected to the extraction, clean-up, and analysis as described previously. The LOD and LOQ were calculated by multiplying the standard deviation of the detected pesticide concentrations from the replicates by 2.99 and 10, respectively. The accuracy was calculated by dividing the recovered concentrations by spiked concentration, and the precision or relative standard deviation (%RSD) of within-laboratory reproducibility analyses, was obtained by dividing the standard deviation by the average concentration. For linearity and calculation of pesticide content in samples, a calibration curve was built using 5 different concentrations of the pesticide standards stock solution (0.1, 0.05, 0.01, 0.005, 0.001 mg/L). 

Obtained linearity was between 0.9987 and 0.9999, recoveries were between 87.8% and 170.8%. This outcome was acceptable according to the European Commission, as the recoveries above the 120% limit were consistently high the 8 in replicate tests [37]. The high recoveries were due to matrix effect. In fact, there is a difference in the signal intensity of an analyte in a matrix solution and its signal in the corresponding solvent. This is due to the presence of matrix impurities that co-elute with the analyte of interest, and interfere with the signal of the detector, resulting in signal enhancement or suppression [38]. The matrix effect has been well documented in pesticide residue analysis using GC-ECD [39,40,41]. In case of signal enhancement, recoveries above 100% will be obtained in a spiking experiment. For recoveries above 120%, calculation adjustments were performed during sample pesticide quantification. The analyses showed good precision, with almost all the %RSD values below the 20% limit. The LODs were between 0.0004 and 0.0652 mg/kg, and the LOQs varied from 0.0012 to 0.2180 mg/kg (Appendix A). The method appeared to be suitable for detecting all the targeted halogenated compounds in the food items [30].

### 2.3. Dietary Exposure and Health Risk Assessment

Human dietary exposure was estimated for determining the degree of risk associated with the detected pesticide residues in food samples. The estimated daily intake (EDI) and estimated short-term intake (ESTI) were calculated and used to determine the chronic or long-term consumer health risk (chronic hazard quotient, cHQ) and the acute or short-term consumer health risk (acute hazard quotient, aHQ). The hazard ratio (HR) was calculated for the carcinogenic risk assessment. 

#### 2.3.1. Deterministic Dietary Exposure

For estimating the dietary exposure to specific residue from each food, daily food consumption rate (in kg/day) was multiplied with the pesticide residue amount in the food (in mg/kg) to obtain the pesticide residue intake. The latter was then divided by the body weight of an adult (kg) to obtain the estimated daily intake, EDI (in mg/kg bw per day).
$$\mathrm{Estimated}\;\mathrm{daily}\;\mathrm{intake}\;\left(\mathrm{EDI}\right)=\frac{\mathrm{Food}\;\mathrm{consumption}\;\times \;\mathrm{Pesticide}\;\mathrm{residue}\;\mathrm{amount}}{\mathrm{Body}\;\mathrm{weight}}$$

On the other hand, because acute poisoning with pesticide from food have been reported in Cameroon [42,43] and very high above MRL values of some pesticides were found in the food samples, the short-term exposure to pesticide residues was assessed. The estimated short-term intake (ESTI in mg/kg bw per day) was calculated by dividing the product of the highest residue value (in mg/kg) and the daily consumption (in kg/person/day) by the body weight (in kg) of an adult [44,45,46].
$$\mathrm{Estimated}\;\mathrm{short-term}\;\mathrm{intake}\;\left(\mathrm{ESTI}\right)=\frac{\mathrm{Food}\;\mathrm{consumption}\;\times \;\mathrm{Highest}\;\mathrm{Pesticide}\;\mathrm{residue}\;\mathrm{amount}}{\mathrm{Body}\;\mathrm{weight}}$$

The residue level used for determining the EDI or the ESTI included aggregation of substances, isomers, metabolites and/or degradation products, performed according to the residue definition for monitoring by European Commission Regulation No. 396/2005 [47]: ƩDDT was sum of *p,p’*-DDT, *op’*-DDT, *pp’*-DDD (dichlorodiphenyl-dichloroethane) and *pp’*-DDE (dichlorodiphenyldichloroethylene); Ʃdieldrin was sum of aldrin and dieldrin; and Ʃendosulfan was the sum of the alpha and beta isomers. Medium percentile exposure was calculated by using the average residue levels, and the 95th percentile exposure was also considered to represent chronic high residue occurrence [48]. 

The major sources of uncertainties in the calculation of dietary exposure to pesticide residues are sampling, analytical bias and variation, left-censored data, body weight if not measured, food consumption levels and food processing factor [49]. The consumption data in this study was obtained from the SSA-TDS, which was established to be used for quantitative risk assessments of food chemicals [35]. Values of food consumption (in g/day) were 81.4 for beans, 2.7 for chili pepper, 36.4 for groundnuts, 384.9 for maize, 1.6 for peas, 1.0 for soybeans, and 1.3 for white pepper. The consumption value of chocolate (0.7 g/day) was used for cocoa, and the value from the ‘’Other Vegetables’’ group (1.4 g/day) was used for Egusi seed. The value for coffee (0.065 g/day) was calculated as the average of the consumption values of North Cameroon (0.1 g/day) and Douala (0.03 g/day). Processing factors, which represent the fraction of the chemical lost from the raw food or agricultural commodity during processing steps were not considered in this study. The average body weight of a Cameroonian adult (64.83 kg) was obtained from [50]. 

The substitution method according to three scenarios was applied [51] for handling the left-censored data (data reported to be below the limit of quantification (LOQ) or non-quantified (NQ)),: the upper bound (UB) which considered NQ = LOQ; medium bound (MB) for which NQ = 1/2 LOQ; and lower bound (LB) for which NQ = zero. Hence, for each pesticide exposure from consumption of each food, the EDI of was assessed at upper, medium and lower bound scenarios and using the average and the 95th percentile of residue content. For the ESTI, only the UB scenario was considered, using the highest value of residue content. 

#### 2.3.2. Non-Carcinogenic Dietary Health Risk Assessment

The chronic or long-term consumer health risk (chronic hazard quotient, cHQ) was calculated by dividing the estimated daily intake (EDI) by the acceptable daily intake (ADI).
$$\mathrm{Chronic}\;\mathrm{Hazard}\;\mathrm{Quotient}\;\left(\mathrm{cHQ}\right)=\frac{\mathrm{Estimated}\;\mathrm{daily}\;\mathrm{intake}}{\mathrm{Acceptable}\;\mathrm{daily}\;\mathrm{intake}}$$

The ADI is an estimate of the daily maximum intake of a substance over a lifetime that will not result in adverse effects at any stage in human life span. The ADI values for pesticides were obtained from the EU Pesticides Database [52] or from the Pesticide Properties DataBase (PPDB) for pesticides not regulated in the European Union, [53]. When HQ < 1, it implies that lifetime consumption of commodity containing the measured level of pesticide residues could not pose health risks. 

The acute or short-term consumer health risk (acute hazard quotient, aHQ) was calculated by dividing the estimated short-term intake (ESTI) by the acute reference dose (ARfD).
$$\mathrm{Acute}\;\mathrm{Hazard}\;\mathrm{Quotient}\;\left(\mathrm{aHQ}\right)=\frac{\mathrm{Estimated}\;\mathrm{short-term}\;\mathrm{intake}}{\mathrm{Acute}\;\mathrm{reference}\;\mathrm{dose}}$$

The reference measure of acute toxicity used was the ARfD, which is an estimate of an oral exposure of a chemical for short-term duration (24 h or less). Values of ARfD were taken from the same sources as the ADI. When the ESTI is less than the ARfD (aHQ < 1), the risk is considered acceptable, and when the ESTI exceeds the ARfD (aHQ > 1), the risk is considered unacceptable. Moreover, the cumulative risk assessment of the combined chronic exposure from a given commodity was performed by using the hazard index (HI) method, which is calculated by summing the cHQs of the individual pesticide residue found in that commodity [54]. A HI < 1 indicates that the concerned commodity should be considered a risk to the consumers, whereas a HI > 1 indicates that its consumption is considered safe [44,45,46]. This helps to understand which commodities are of most concern of health risk due to pesticide residue contamination.

#### 2.3.3. Carcinogenic Dietary Risk Assessment

Carcinogenic risk estimates are expressed in terms of the probability that an individual will contract cancer over a lifetime of exposure to the pesticide residue. For the carcinogenic risk assessment, the hazard ratio (HR) was calculated by dividing the EDI by the cancer benchmark concentration (CBC). The CBC for carcinogenic effect is derived by setting the risk to 1 in 1,000,000 due to a lifetime exposure to the pesticide residue, and was calculated using the formula: $$\mathrm{CBC}=\frac{\mathrm{RL}\;\times \;\mathrm{BW}\;}{\mathrm{CR}\;\times \;\mathrm{OSF}}$$
where RL is the maximum acceptable risk level (1 × 10^−6^) and represents the increased probability of developing cancer over the lifetime as a result of exposure to the pesticide residue, BW is the body weight (kg), CR is the food consumption rate (g/day), and OSF is the oral slope factor (mg/kg/d) [55]. Oral slope factor values were obtained from the Integrated risk information system (IRIS) [56]. When HR exceeds 1, it indicates that there is potential risk to human health.

### 2.4. Determination of the Source of DDT Residues in the Food Items 

The method of [57] was used to establish whether DDT residues found in foods come from historical soil persistence of from recent application. Specific ratios of isomers and parent/metabolites of OCP compounds have been widely used to monitor exposure time, metabolism, storage of compounds, and the source of contamination (past and present application) in different matrices. The method initially used for soil and water samples is also used for food samples, for examples: fruits and vegetables [58], Khat [59], milk, [60], chicken products [61] and fish [62]. Degradation of p,p′-DDT under aerobic and anaerobic conditions generates p,p′-DDE and p,p′-DDD metabolites, respectively. The ratio of concentration of the parent p,p′-DDT to its metabolites (p,p′-DDD and p,p′-DDE) is used for assessing the possible pollution sources: if the ratio of metabolite/parent is found to be higher than 1, it indicates historic presence of DDT compounds, while a ratio lower than 1 indicates recent application of p,p′-DDT. 

### 2.5. Data Analysis 

Descriptive statistics were generated. The compliance of quantified pesticides with existing regulations was checked by comparing their level with European Union MRLs obtained from the EU Pesticides database [52]. The proportions of occurrence in foods, above the MRL and in each sampling location were computed for each pesticide. The residue content was expressed as lowest, highest, mean and median. The proportions of positive analyses, quantified residues, analyses above MRL, and pesticide compound above MRL were determined for each food item. The significance of the difference in HI between the lower, medium and upper bound scenarios was determined by two-tailed F-test at 0.05. The major quantified pesticide residues contributing to the hazard index of each commodity was determined and graphically presented: pesticides with values of relative contribution lower than 1% were grouped as ‘’others’’. The data analysis function of Microsoft Excel 2019 was used. 

## 3. Results and Discussion

### 3.1. Concentration of Halogenated Pesticides in Foods from Cameroon

#### 3.1.1. Distribution of Pesticide Residues Found in Food Samples

Residues of 20 halogenated pesticides were assessed by GC-ECD in 11 highly consumed agricultural products from the three largest cities of Cameroon (Appendix A). Figure 1 shows examples of GC-ECD chromatograms of a standard and a food sample. All the 20 compounds were detected and quantified in the 160 samples (Table 1). Organochlorine was the most detected pesticide class (found in 5.6 to 85.0% of sample), followed by phtalimides (68.8%), chloronitriles (66.3%) and pyrethroids (18.1–51.9%), whereas phenylamides were found in 21.9% of samples. Fungicides were detected in 61.9–68.8% of samples, insecticides in 5.6 to 85.0%, and the herbicide alachlor in 21.9%. In our preliminary work on the samples of same food items originating from the western highlands of Cameroon [30], all the tested halogenated pesticides were also found in the food items. This confirms a high occurrence of residues of halogenated compounds in food items from Cameroon. 

The most frequently detected individual pesticides were aldrin, p,p’-DDT and β-HCH which were found in 85.0%, 81.9% and 72.5% of samples, respectively. Conversely, β-Endosulfan could be quantified in only 5.6% of samples. Aldrin, β-HCH and bifenthrin were positive in all the 11 food items, while β-endosulfan was found in only 4 food items, namely coffee, Egusi seeds, kidney beans and maize. Regarding the sums of compounds, ƩDDT and Ʃdieldrin were quantified in 95.6% and 86.9% of samples, respectively, and in all the 11 food items, while Ʃendosulfan was in 49.1% of samples of 10 food items. The distribution of the quantified pesticides among the 3 sampling locations revealed that all the 20 pesticides were found in samples from each location. The highest contaminations found were methoxychlor in chilli pepper from Douala (7.25 mg/kg), alachlor in chilli pepper from Bafoussam (5.81 mg/kg), cypermethrin in maize from Yaounde (1.84 mg/kg) and β-HCH in chilli pepper from Bafoussam (1.56 mg/kg). However, on average, methoxychlor (1.12 mg/kg), alachlor (0.74 mg/kg) and β-HCH (0.39 mg/kg) were the compounds found with the highest concentration, all in chilli pepper. In the opposite, DDT metabolites o,p’-DDT and p,p’-DDE showed the lowest average concentrations (0.01 mg/kg). In general, most of the highest residue content were found in chilli pepper, followed by maize and white pepper. 

The most distributed pesticides and the proportion of positive samples of the results obtained here show some similarities with the findings of our previous work. However, for almost all the compounds, the level of contamination obtained in this study are higher than the previous results, where HCB at 3.09 mg/kg was the highest residue level found, followed by cypermethrin at 0.94 mg/kg. Moreover, the range of average contamination levels in this study (0.01–0.58 mg/kg) is higher than the previous study (<0.01–0.18 mg/kg) [30]. In other comparable studies in Cameroon, organochlorine pesticides were detected with high frequency and in high concentration, ranging from 0.02 mg/kg for β-endosulfan in millet to 9.53 mg/kg lindane (γ-HCH) in maize, and permethrin was found in maize at 0.39 mg/kg [23]. Likewise, cypermethrin was found in leafy vegetables from Douala (0.23 mg/kg) and Garoua (0.13 mg/kg) [32]. 

Halogenated pesticides were found in food samples from other countries. In a TDS in Benin, Cameroon, Mali and Nigeria, the most frequently found compounds were pyrethroids (35.7%), chlorpyrifos (22.4%), cypermethrin (18.0%), dichlorvos (13.6%), lambda cyhalothrin (8.2%), permethrin (7.5%) and profenofos (5.8%) [32]. In Nigeria, residues of 1 or more OCPs were quantified in approximately 96% of the analyzed maize samples [63], and in 17 food items from Nigerian Markets, OCPs were generally low, with p,p’-DDE having the highest residue concentrations of 0.11 mg/kg in pulses and 0.12 mg/kg Cameroon peppers and chili peppers [64]. The concentrations of OCPs in fruits (watermelon, pineapple, and banana) from Ghana ranged from not detectable to 0.05 mg/kg for DDTs, 0.02 mg/kg for HCHs, 0.004  mg/kg for chlordanes, 0.02  mg/kg for aldrin, and 0.01  mg/kg for other OCPs [65]. In another study on residues of 13 halogenated pesticides from 130 fruits samples in Accra Ghana, residue levels varied from 0.01 to 0.26 mg/kg [66]. Several studies reported residues of halogenated pesticides in foods from Ethiopia [67,68,69]. Lower residue concentrations were found in the 2007 Chinese total diet study [70] and in vegetables from Indian markets [58]. The higher contamination rates obtained in our study and other studies in Cameroon, as compared to reports from other countries, raise the concern of the usage and regulation of these pesticides in Cameroon. 

#### 3.1.2. Compliance of the Quantified Pesticides with Regulation Limits 

All the tested compounds showed residues above their existing EU MRLs in one or more samples (Appendix A), and in general, 40.05% of the positive analyses were above the MRLs. All the 35 positive samples of alachlor had their residue levels above the MRLs, followed by methoxychlor (88.2%) and β-HCH (78.4%). On the other hand, deltamethrin, Ʃendosulfan and cypermethrin showed only 6.9%, 6.0% and 5.7% of their residues above MRLs, respectively. All the 14 pesticides banned in Cameroon analyzed in this study, were found in the food samples, and all of them had some residues above the existing MRLs, with alachlor showing 100% of positive samples above the MRLs (Table 1, Figure 2). 

In our previous findings, only 34.4% of the pesticides were found above their existing MRL values and in 38% of the positive analyses. Further, only 11 of the halogenated compounds had above MRLs residues [30]. Similarly, another study found that in maize, cowpea and millet from northern Cameroon, 75% of samples containing pesticide residues above MRLs were found [23]. The high level of samples above the compliance limit observed in this study can be the result of insufficient training and deficient assistance of farmers from agricultural extension agents that leads to lack of Good Agricultural Practices (GAP), with inappropriate applications of pesticides by farmers, [15,16]. This highlights the necessity of regulatory authorities in Cameroon, to take more action to regulate usage of agrochemicals in the country [26,30,31]. 

The banned compounds found in this study were prohibited in the country many decades ago: 31 years for dieldrin, aldrin and heptachlor, 15 years for lindane (γ-HCH), 12 years for endosulfan [36]. Almost all of these pesticides are POPs and therefore, presence of their residues in foods today can be attributed to environmental persistence. Obsolete pesticides have been reported as one of the major problems in developing countries, where the OCPs are often detected in different food items mainly as a result of environmental contamination from previous applications for agricultural and anti-malarial purposes, and leaching from dumped obsolete stocks [71]. However, the levels and frequencies of residues of banned compounds recorded in this study are higher than in many other reports. For instance, residue levels of OCPs from the Chinese TDS were significantly below the extraneous maximum residue limits (EMRLs) [70]. In vegetables from India, 65.4% of samples exceeded the prescribed MRL for heptachlor, aldrin exceeded the MRL in 30.8% of samples, chlordanes in 28.8%, endrin in 5.8%, and methoxychlor in 3.9% of samples, whereas MRLs for ΣDDT and Σendosulfan were not exceeded in any samples [58]. In Ghana, 32.8% of the fruit samples analyzed contained residues of the monitored halogenated pesticides above the accepted MRLs [66], in watermelon, levels of methoxychlor, Aldrin and γ-HCH exceeded the MRLs [64]. Only one-third of samples of maize, teff, red pepper and coffee from Ethiopia had residues above the limits [68] although in other study, all the 127 maize samples contained residues of DDT and its metabolites above the MRL [67]. 

#### 3.1.3. Source of DDT Contamination in Food Samples 

Out of the 131 samples in which residues of DDT and its metabolites were found and computation of metabolite/parent ratio was possible, p,p′-DDD/p,p′-DDT was >1 in only 3 samples all of chili pepper, and for the 128 remaining samples (97.71%), the ratio was <1. Similarly the ratio p,p′-DDE/p,p′-DDT was >1 in only 6 samples all of chili pepper, and a ratio was <1 was recorded for the other 125 (95.42%) samples. Abundance of parent compound over the metabolites indicates recent application of DDT [57], and confirms the illegal use of this obsolete pesticide in Cameroon. In fact, the FAO in 1996 had recorded a stock of 225,000 kg of unwanted POPs in 20 sites in Cameroon [27], and later, important stocks of obsolete pesticides accumulated over the years (>200,000 kg and 300,000 L) were inventoried in the country [26]. Because these obsolete pesticides are still effective against pathogens and pests, and are cheap on the black market, aided by limited control by the authority, they can be illegally used on crops and produces [29]. Similar evidence of presence of DDT over its degradation products was reported in India, were almost 54% of samples showed a ratio of p,p′-DDD/p,p′-DDT <1 [58]. In southwestern Ethiopia however, the presence of higher concentration of the metabolite p’p-DDE (0.03–0.11 mg/kg) compared to the parent compound p’p-DDT (0.010–0.026 mg/kg) in khat (*Catha edulis*) indicated the historical use of DDT in the study area [59]. However, in our study, the opposite was observed, with parent p,p’-DDT having higher frequency and residue levels than metabolites p,p′-DDD and p,p’-DDE. This demonstrates that Cameroonian authority needs to strengthen the controls of stock of obsolete pesticides and effective implementation of pesticide bans in the country. 

#### 3.1.4. Contamination of the Food Items with Pesticide Residues 

All the 11 food items were contaminated with pesticides (Appendix A). Chili pepper (58.9% of positive analyses), cowpea (56.8%), black beans (56.5%) and kidney beans (54.0%) were the most contaminated food items, showing positive for more than half of the analyses. With only 20.7% of the samples tested positive, cocoa was the least contaminated item. In terms of individual pesticides, out of the 20 tested, chili pepper, black beans and maize contained 19 pesticides each, while 10 pesticides were found in soybeans. Regarding the compliance of the food items with EU regulatory limits, 40.4% of chili pepper analyses and 25.3% of maize analyses were found above the MRLs, whereas cocoa was the most compliant food item, with only 7.0% of analyses above the MRLs. Chili pepper and maize contained 17 (85.0%) and 13 (65.0%) pesticides, respectively, exceeding the MRLs, while soybeans contained only 6 (30.0%) (Figure 3). Concentration-wise, chili pepper showed the highest average concentration, followed by white pepper and maize (Figure 4). In the previous study, chili pepper, white pepper, kidney beans and soybeans were the food items with the highest number of residues, while kidney beans, soybeans, chili pepper, and maize were found with the highest residue concentrations [30]. 

#### 3.1.5. Distribution of Pesticide Residues among the Sampling Locations 

There was no significant difference in the distribution of contaminated food samples among the 3 sampling cities (Appendix A). In Bafoussam, Douala and Yaounde, 33.5%, 34.2% and 32.3% of the analyses were positive, respectively. Likewise, respectively 32.7%, 34.3% and 33.0% of positive samples in Bafoussam, Douala and Yaounde contained residues above the MRLs. These results do not agree with a recent study in which the contamination of composite foods from Cameroon with organophosphate residues varied with the location (Douala and Garoua), and with the season of sample collection (rainy and dry season). Dichlorvos was detected in 8% and 7% of food samples from Douala and Garoua. Cypermethrin was quantified in leafy vegetables collected during the dry season in Douala (0.235 mg/kg) and in Garoua (0.126 mg/kg) but was not detected in vegetables collected during the wet season. Beans from Garoua, smoked fish from Garoua and from Douala, and vegetables from Douala contained 0.475, 0.015, 0.012 and 0.006 mg/kg of Permethrin, respectively. Chlorpyrifos was found in two composites samples of tomato from Douala (at 0.026 and 0.091 mg/kg), in leafy vegetables from Garoua (0.033 mg/kg) and in bread from Douala (0.014 mg/kg) [32]. 

These results reflect the poor knowledge of farmers on pesticides and in practices of pesticide usage in the different parts of the country, demonstrated in previous reports [14,16,17,21,22,23,24,25]. Our findings also indicate that the usage of halogenated pesticides in Cameroon may not vary in the different agroecology areas of the countries. In fact, Bafoussam is located in western highlands agroecological zone of Cameroon, while Douala is in the humid forest with monomodal rainfall zone, and Yaounde in the humid forest with bimodal rainfall zone. These different agroecology should have imposed different agricultural constrains that would necessitate different patterns of pesticide usage in each area. 

### 3.2. Carcinogenic and Non-Carcinogenic Health Risk Assessment 

#### 3.2.1. Dietary Chronic Risk Assessment of Pesticide Residues 

The ADI was available and hence the chronic risk assessed for only 13 compounds. When assessing the chronic health risk using the average value of residue level, maize appeared to be unsafe for 2 compounds, with the cHQs above 1 for Ʃdieldrin (2.89 under the 3 scenarios) and for heptachlor (1.89, 1.53 and 1.17 under the UB, MB and LB scenarios, respectively). Additionally, endrin in maize, heptachlor in black beans, and Ʃdieldrin in black beans, kidney beans and groundnuts showed cHQ values between 0.65 and 0.13: although they can be considered safe, these compounds in the indicated foods should be watched (Table 2). Chronic risk assessment using the 95th percentile of residue level showed that the cHQ values above 1 were obtained in maize for 3 residues Ʃdieldrin (cHQ = 11.72), heptachlor (cHQ = 5.67) and endrin (cHQ = 1.21), and for Ʃdieldrin in black beans (cHQ = 1.91) and Ʃdieldrin in kidney beans (cHQ = 1.45). These values were equal under the 3 scenarios. Moreover, cHQ between 0.58 and 0.10 were obtained for heptachlor, Ʃdieldrin, cyhalothrin, deltamethrin and bifentrhin in black beans, groundnuts, maize, kidney beans and chili pepper. For all the other foods and compounds, cHQ values were below 0.10 (Appendix A).

Our results are higher than the previous report in Yaounde Cameroon [28], which reported HQ from 0.0024 (cypermethrin) to 0.0303 (pirimiphos-methyl) using the average exposure, and 0.0042 and 0.0514 using the 95th percentile exposure, respectively, all under the UB scenario. This suggests that for the pesticides analyzed in these studies, during the last 12 years, the majority of foods consumed in large cities of Cameroon became more likely to cause health problems. Studies from other countries strengthen this suggestion, and highlight comparatively very high health risk in Cameroonian foods. In food items from Nigerian markets, the risk index was <1 in most cases, with the exception of p,p’-DDD in fruits for children [64]. In fruits from Ghana, aldrin in watermelon could pose potential toxicity to the consumer [65]. Risk assessment of DDT and its metabolites in maize suggested that the use of maize as complementary food for infants in Ethiopia may pose a health risk [67]. In a total diet study on pesticide residues in France, exposure levels were below the ADI for 90% of the pesticides under the UB and LB scenarios [72]. Risk assessment showed that serious public health problem could not be considered for similar pesticides in foods in Seoul, Korea [34], Turkey [73], Lebanon [45] and China [70], while aldrin and heptachlor epoxide could be considered as a serious concern for Delhi population in India [58]. 

#### 3.2.2. Dietary Acute Risk Assessment of Pesticide Residues 

The aHQ was calculated only for 8 compounds with existing ARfD values. All the aHQ values were below 1 and therefore, the risk is considered acceptable, indicating that incidental cases where residue levels in the analyzed food items may potentially pose acute toxicity threat to Cameroonian adult consumers could be excluded. The highest aHQ values were both in maize, for Ʃdieldrin (0.588) and deltamethrin (0.355), suggesting that maize is a commodity that should be monitored for acute toxicity of pesticide residues (Appendix A). In the Greek population, the values of aHQ in samples of grapes and olive oil ranged from 0.01 to 0.82, acute risks from oxamyl in peppers (aHQ = 0.54), tomatoes (aHQ = 0.51) and cucumbers (aHQ = 0.24) were considered high. The authors pointed out some possibility of acute risk for the consumers and highlighted the necessity of continuous monitoring [74]. 

#### 3.2.3. Cumulative Dietary Risk Assessment of Multiple Pesticide Residues in Each Food Item

The HI values representing chronic health risk of cumulative pesticides in each food item at medium and high percentile exposures, considering the 3 scenarios of residue estimation, are compiled in Table 2. At the average residue level, the risk was high for maize (HI between 4.78 and 5.75) indicating that there is a health risk for Cameroonian adults consuming maize. At the 95th percentile exposure HI valued showed between 1.70 and 1.72 for kidney beans, 2.58 and 2.60 for black beans, and values as high as 19.42 to 19.46 for maize. These results show that for high-end Cameroonian adult consumers, the risk of cumulative chronic health risk cannot be excluded for beans, and there is a very high risk for consuming maize. 

Hazard indexes of vegetables contaminated with OCPs in India were 2.6 and 12.2 for adults and children respectively, which exceeded the recommended guideline value and revealed a serious health concern for these populations [58]. In a study of risk of organochlorine pesticides from fruits in Ghana, there was a high combined health risk of consumption of watermelon from Kyerefamso in children (HI = 2.71 at average level of exposure). No cumulative risk associated with the consumption of other fruits from different towns in both adults and children was recorded [65]. The HI at average level of exposure for all residues in apple form Lebanon was 0.19, indicating that there is no risk of adverse effects following a cumulative exposure to all the detected pesticides [45]. 

In a study of Greek population dietary chronic exposure to pesticide residues in fruits, vegetables and olive oil, pyrethrins and organochlorine (HI value of 0.052 and 0.087, respectively) presented a negligible hazard for the consumers. Organophoshates and carbamates (HI = 0.389 and 0.343, respectively) did not constitute a risk but required further attention. The authors recommended a long term monitoring program from different areas and different times of sampling for more accurate conclusions [74]. Similarly, our results show that at average exposure level, HI of black beans (0.69–0.79) and kidney beans (0.46–0.57), although not presenting a risk, necessitates further attention. The same observation is valid for high-end exposure level of chili pepper (HI = 0.31), and groundnuts (HI between 0.26 and 0.31). In fact, many food items analyzed in this work make up the majority of dishes consumed by urban and rural Cameroonian populations, which are based on tubers, cereals and legumes, most of which are consumed as composite meals [75,76,77,78]. Considering this, and without taking into account the processing factor, the cumulative risk of composite meals may show that they are not safe for consumers. 

The hazard index is calculated by summing the hazard quotient of the individual pesticide residues found in a commodity. It is the most useful methods of cumulative risk assessment of pesticide residues in food, additionally offering a low level of complexity and refinement. The limitation is that since this method is based on an assumption of additivity it can lead to errors if a synergistic or antagonistic action occurs. There are more refined but complex and less used methods like the reference point index, the relative potency factor method and physiologically based toxicokinetic modelling [44,54].

#### 3.2.4. Major Contributor Pesticides to Cumulative Chronic Health Risk

The average exposure level at the medium bound scenario was considered and only critical commodities (maize, kidney beans and black beans), and those who require further monitoring attention (chili pepper and groundnuts) are presented here. Results show that for these commodities, 6 compounds essentially contribute to the HI: Ʃdieldrin was the highest major contributor, accounting for 48% to 77% of the HI, followed by heptachlor (11% to 41%), endrin (4% to 11%), alachlor (1% to 3%), ƩDDT (1% to 2%) and cyhalothrin (1% to 2%) (Figure 5). Propamocarb and chlorpyrifos were the major contributors to HI for green pepper and cucumber in Turkey, but the low HI values indicated no concern of cumulative exposure [73]. The major contributor OCPs to non-carcinogenic chronic health risk from fish consumption in Africa were aldrin (50% for freshwater fish and 59% for marine/saltwater fish), dieldrin (35% and 25%), ∑HCH (9% for freshwater fish) and DDT (15% for marine/saltwater fish) [79]. When establishing regulation measures for reducing the health risk, the most concerned pesticide(s) for each food can be targeted for more effective policy implementation. 

#### 3.2.5. Cancer Risk Assessment 

The HR was calculated for the 9 pesticides with existing OSF values in the IRIS database and the carcinogenic risk was assessed at medium and high end exposure level, under 3 scenarios (Appendix A). For the 95th percentile exposure, there was no significant difference (F-test, *p* < 0.05) between the HR values for the 3 scenarios, therefore, only the upper bound values are presented in Table 3, which depicts the results for HR > 1. It appears that all the 9 assessed pesticides exhibited concerned risk of carcinogenicity, in 5 food items, namely maize, black beans, kidney beans, groundnuts and chili pepper. For all the pesticides, very high HR values were obtained in maize, suggesting a very high risk of cancer from consumption of this food item. For instance, HR value of dieldrin in maize was between 16,038.25 and 15,850.25 at average exposure, and 58,395.59 at high end exposure. This means that, there is approximately a 16,000 in 1,000,000 chance of getting cancer from dieldrin for an average maize consumer, the chance rises to 58,000 in 1,000,000 for a high maize consumer in Cameroon.

In Ghana, HR for p,p’-DDT in watermelon was >1, indicating that its contamination could pose potential carcinogenic risk in children, whereas the risk of OCP residues in the other fruits was of less concern [65]. From the Chinese TDS, 5 OCPs showed HR values above 1 [70]. In the Indian study, HR values were observed in the order of 627 to 1.47 in adult, and 922 to 2.85 in children, depicting potential risk of cancer via consumption of these contaminated vegetables [58]. Values of cancer risk as high as 3500 were found for OCP residues in fish samples collected from Lake Chad [80], and OCPs in fish samples from different African countries showed values higher than the permissible limit, indicating possible development of cancer through fish consumption [79]. It can be observed that cancer risk form the majority these studies are very lower than the values obtained in this study in Cameroon. A recent study has found the presence of malathion, parathion and chlorpyrifos residues in Cameroonian blood samples, and this chronic exposure to a mixture of pesticides lead to several metabolic dysregulations which may result in the establishment of metabolic syndrome and other chronic diseases [81].

## 4. Conclusions and Perspectives

In this first study of determination and risk assessment of halogenated pesticide residues in Cameroon, all the 20 halogenated compounds analyzed were detected and quantified in the 160 samples of 11 agricultural products, fungicides (61.9–68.8% occurrence) were the most found group, organochlorines (5.6 to 85.0%) were the most represented chemical group, and aldrin, p,p’-DDT and β-HCH (85.0%, 81.9% and 72.5%, respectively), were the most distributed individual pesticides. Methoxychlor, alachlor and β-HCH (1.12, 0.74 and 0.39 mg/kg, respectively) in chilli pepper were found with the highest average concentrations, while residue level in an individual sample as high as 7.25 mg/kg methoxychlor was found in a chilli pepper sample from Douala. Most of the highest residue content were found in chilli pepper, maize and white pepper. All the 20 compounds showed residues above their existing EU MRLs in one or more samples and 40.05% of positive samples, including all the 35 samples containing alachlor, exceeded the regulatory limits, suggesting a risk for the consumers and limiting the opportunities of exportation to the European market. All the 14 pesticides banned in Cameroon analyzed in this study were found in the food samples, and all showed residues above the MRLs. Parent DDT was more abundant than its metabolites in 95.42% to 97.71% of samples, indicating recent application of DDT on these foods or their crops, and confirming the illegal use of this obsolete pesticide in Cameroon. All the 11 food items were contaminated with pesticides, with chili pepper (58.9% of positive analyses), cowpea (56.8%), black beans (56.5%) and kidney beans (54.0%) showing the highest pesticide occurrence, and the distribution of contaminated samples among the 3 sampling cities was similar.

Risk assessment showed that Ʃdieldrin, heptachlor, and endrin in maize, black beans, and kidney beans could be considered unsafe in term of chronic health concern to consumers. The acute risk was considered acceptable for the 8 compounds assessed. Cumulative health risk showed that maize, kidney beans and black beans could pose health problems because of the heavier contribution compared with other food items in the cumulative pesticide toxicity. For all these commodities with health risk concern, 6 compounds (Ʃdieldrin, heptachlor, endrin, alachlor, ƩDDT and cyhalothrin) essentially contributed to the hazard index. Nine assessed pesticides exhibited seriously concerned risk of carcinogenicity in 5 food items (maize, black beans, kidney beans, groundnuts and chili pepper), with the risk being very high for dieldrin in maize.

Comparative to other studies, the very high values of residues and risk parameters obtained in this study can be justified by the probable application of storage pesticides on these foods, and the high consumption rate (as for maize). Additionally, the dry nature of all the food products analyzed in this study can be another reason, as dry items usually show higher concentration of compounds. Assessing the waiting period of these pesticides, and the effect of washing and different local cooking practices on their residue content will provide a more precise understanding of the health risk they can pose to consumers. In the meantime, the outcomes of this study can be used when designing future pesticide control programs to minimize human health risks in Cameroon. In fact, our results indicate that Cameroonian regulatory authorities must control the usage of agrochemicals in the country, strengthen the measures for effective implementation of the pesticide bans and manage the huge stocks of obsolete pesticides currently present in the country. These measures are of paramount importance not only for reducing the growing health risk for Cameroonian populations, but also to guarantee approval of Cameroon export produces on the international market.

## Figures and Tables

**Figure 1 ijerph-18-05043-f001:**
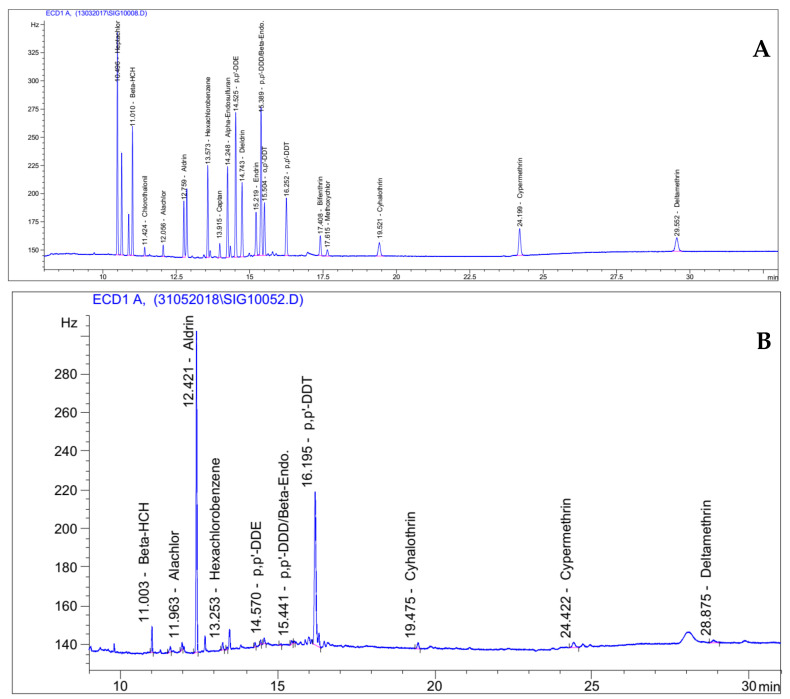
Examples of gas chromatography-electron capture detector (GC-ECD) chormatograms for the standards at 0.01 mg/L (**A**) and a kidney bean sample (**B**).

**Figure 2 ijerph-18-05043-f002:**
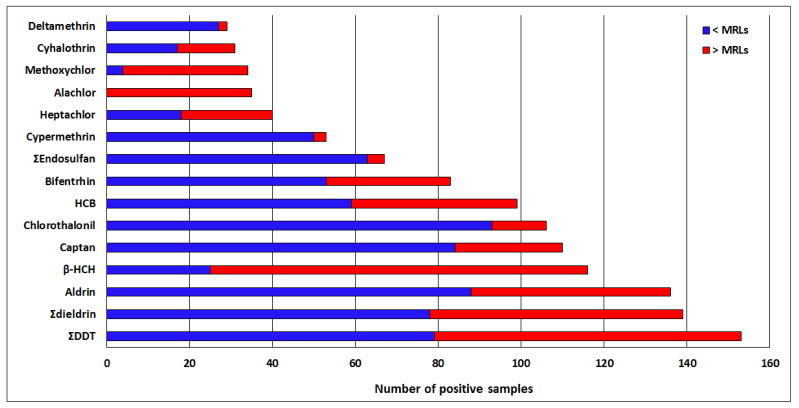
Frequency of positive analyses of regulated halogenated pesticides residues in 11 food items from 3 cities of Cameroon, with indication of number of samples below and above the European Union maximum residue limits (MRLs). HCB = hexachlorobenzene, HCH = hexachlorocyclohexane, DDT = dichlorodiphenyltrichloroethane. ƩDDT = p,p’-DDT + op’-DDT + pp’-DDD + pp’-DDE. Ʃdieldrin = aldrin + dieldrin. Ʃendosulfan= α-endosulfan + β-endosulfan.

**Figure 3 ijerph-18-05043-f003:**
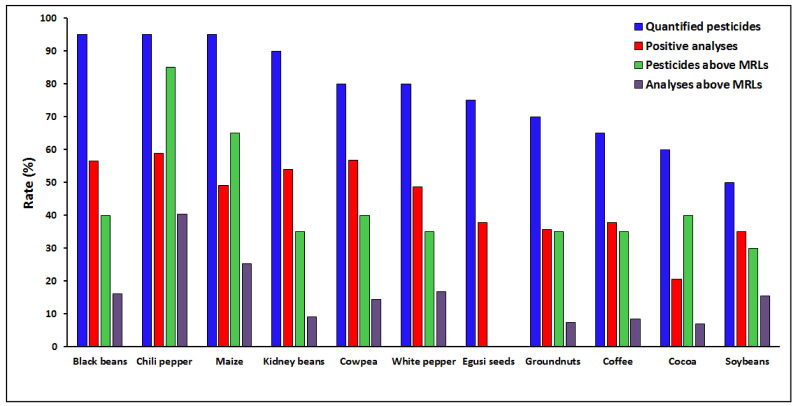
Representation of quantified halogenated pesticide residues among 11 food items from the 3 largest cities of Cameroon.

**Figure 4 ijerph-18-05043-f004:**
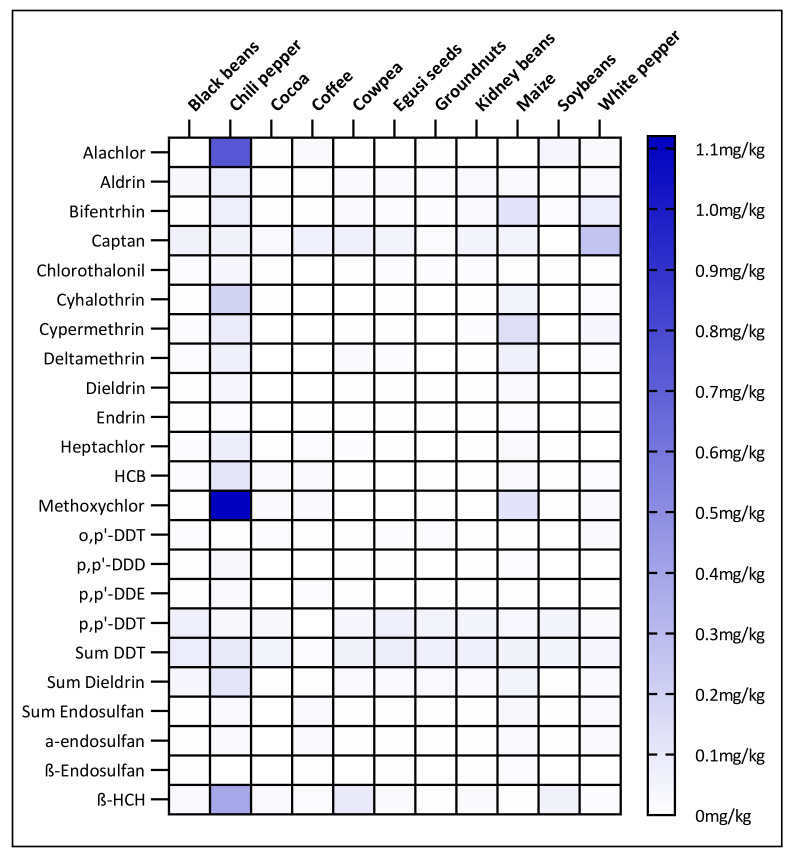
Average concentrations of halogenated pesticide residues in 11 food items from the 3 largest cities of Cameroon. HCB = hexachlorobenzene, HCH = hexachlorocyclohexane, DEE = dichlorodiphenyldichloroethylene, DDD = dichlorodiphenyldichloroethane, DDT = dichlorodiphenyltrichloroethane. ƩDDT = p,p’-DDT + op’-DDT + pp’-DDD + pp’-DDE. Ʃdieldrin = aldrin + dieldrin. Ʃendosulfan= α-endosulfan + β-endosulfan.

**Figure 5 ijerph-18-05043-f005:**
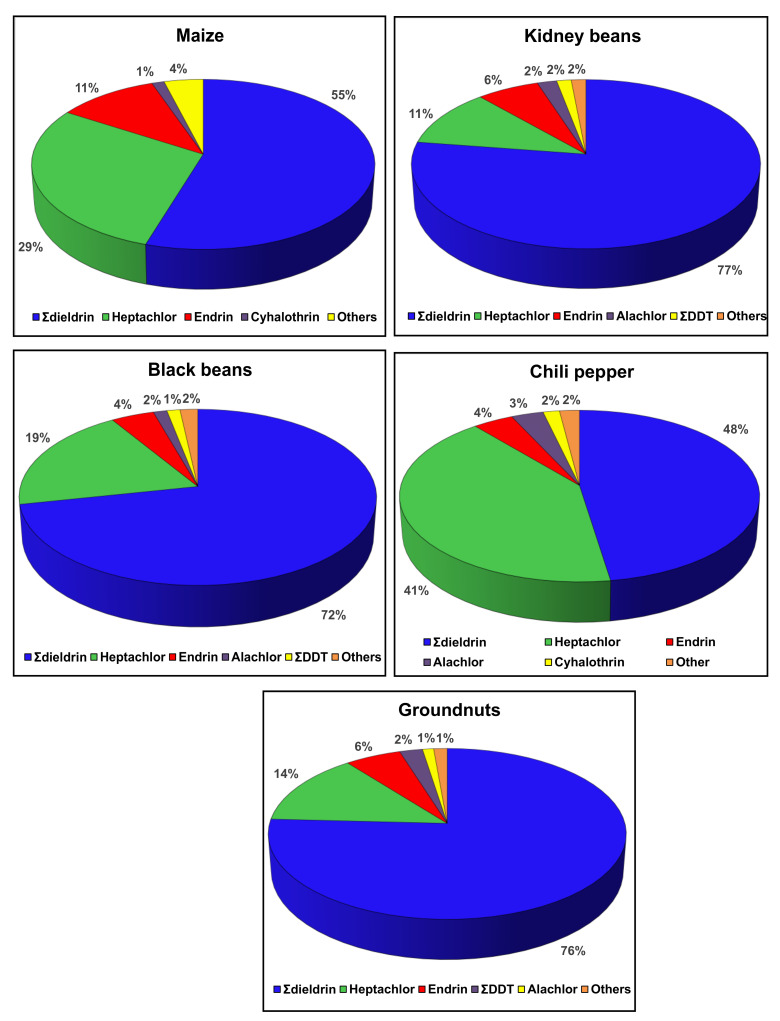
Relative contribution of halogenated pesticide compounds to the cumulative chronic health risk of food items from the 3 largest cities of Cameroon. ƩDDT = p,p’-DDT + op’-DDT + pp’-DDD + pp’-DDE. Ʃdieldrin = aldrin + dieldrin.

**Table 1 ijerph-18-05043-t001:** Distribution of halogenated pesticide residues quantified in 11 food items from the 3 largest cities of Cameroon.

Sr. No.	Pesticide Residue	Chemical Class	IARC Group	Application	Banned in Cameroon?	Number of Positive Samples	Percent of Positive Samples (%)	Number of Positive Food Items	Lowest Residue Value (mg/kg)	Highest Residue Value (mg/kg)	Mean Residue Value (mg/kg)	Median Residue Value (mg/kg)	95th Percentile (mg/kg)	Number of Samples >MRLs	Percent of Positive Samples > MRLs (%)
1	Alachlor	Phenylamides		Herbicide	Yes	35	21.9	6	0.0240	5.8116	0.3532	0.0684	1.0893	35	100.0
2	Aldrin	Organochlorines	2A	Insecticide	Yes	136	85.0	11	0.0051	0.3240	0.0291	0.0144	0.1131	48	35.3
3	Bifenthrin	Pyrethroid		Insecticide	No	83	51.9	11	0.0072	0.5691	0.0674	0.0246	0.2457	30	36.1
4	Captan	Phtalimides	3	Fungicide	No	110	68.8	10	0.0345	0.4818	0.0897	0.0522	0.3309	26	23.6
5	Chlorothalonil	Chloronitrile	2B	Fungicide	No	106	66.3	10	0.0057	0.2550	0.0165	0.0101	0.0469	13	12.3
6	Cyhalothrin	Pyrethroids		Insecticide	No	31	19.4	7	0.0111	0.8235	0.1258	0.0756	0.4931	14	45.2
7	Cypermethrin	Pyrethroids		Insecticide	No	53	33.1	7	0.0114	1.8439	0.0885	0.0312	0.2053	3	5.7
8	Deltamethrin	Pyrethroids	3	Insecticide	No	29	18.1	7	0.0117	0.5979	0.0964	0.0370	0.3343	2	6.9
9	Dieldrin	Organochlorines	2A	Insecticide	Yes	54	33.8	7	0.0033	0.2871	0.0247	0.0083	0.0986	NA	NA
10	Endrin	Organochlorines	3	Insecticide	No	42	26.3	8	0.0039	0.0540	0.0190	0.0134	0.0458	NA	NA
11	Heptachlor	Organochlorines	2B	Insecticide	Yes	40	25.0	9	0.0048	0.3993	0.0544	0.0117	0.2810	22	55.0
12	HCB	Organochlorines	2B	Fungicide	No	99	61.9	10	0.0051	0.8904	0.0330	0.0090	0.1162	40	40.4
13	Methoxychlor	Organochlorines	3	Insecticide	No	34	21.3	8	0.0351	7.2504	0.5823	0.2064	1.8143	30	88.2
14	o,p’-DDT	Organochlorines	2A	Insecticide	Yes	89	55.6	10	0.0078	0.0366	0.0143	0.0120	0.0329	NA	NA
15	p,p’-DDD	Organochlorines		Insecticide	Yes	24	15.0	7	0.0032	0.2193	0.0322	0.0105	0.1818	NA	NA
16	p,p’-DDE	Organochlorines		Insecticide	Yes	80	50.0	8	0.0039	0.1392	0.0113	0.0060	0.0276	NA	NA
17	p,p’-DDT	Organochlorines	2A	Insecticide	Yes	131	81.9	10	0.0087	0.1770	0.0507	0.0486	0.0964	NA	NA
18	α-endosulfan	Organochlorines		Insecticide	Yes	63	39.4	10	0.0060	0.1617	0.0287	0.0233	0.0641	NA	NA
19	β-Endosulfan	Organochlorines		Insecticide	Yes	9	5.6	5	0.0045	0.0408	0.0178	0.0141	0.0370	NA	NA
20	β-HCH	Organochlorines	2A	Insecticide	Yes	116	72.5	11	0.0048	1.5567	0.0864	0.0372	0.2921	91	78.4
21	*Ʃdieldrin*	Organochlorines		Insecticide	Yes	139	86.9	11	0.0045	0.3552	0.0381	0.0198	0.1442	61	43.9
22	*ƩDDT*	Organochlorines		Insecticide	Yes	153	95.6	11	0.0063	0.3364	0.0627	0.0573	0.1251	74	48.4
23	*ƩEndosulfan*	Organochlorines		Insecticide	Yes	67	41.9	10	0.006	0.185	0.030	0.024	0.064	4	6.0

HCB = hexachlorobenzene, HCH = hexachlorocyclohexane, DEE = dichlorodiphenyldichloroethylene, DDD = dichlorodiphenyldichloroethane, DDT = dichlorodiphenyltrichloroethane. ƩDDT = p,p’-DDT + op’-DDT + pp’-DDD + pp’-DDE. Ʃdieldrin = aldrin + dieldrin. Ʃendosulfan= α-endosulfan + β-endosulfan. NA = Not applicable, because it doesn’t exist, maximum residue limit of individual compounds.

**Table 2 ijerph-18-05043-t002:** Hazard index values representing chronic health risk of cumulative pesticides in 11 food items from the 3 largest cities of Cameroon.

Sr.No	Food Item	Hazard Index					
		Average Exposure			95th Percentile Exposure		
		Upper Bound	Medium Bound	Lower Bound	Upper Bound	Medium Bound	Lower Bound
1	Black beans	0.7958	0.7433	0.6908	**2.6025**	**2.5918**	**2.5811**
2	Chili pepper	0.1052	0.0966	0.0880	0.3053	0.3053	0.3053
3	Cocoa	0.0068	0.0040	0.0011	0.0095	0.0072	0.0050
4	Coffee	0.0005	0.0004	0.0003	0.0012	0.0012	0.0011
5	Cowpea	0.0109	0.0102	0.0096	0.0324	0.0322	0.0319
6	Egusi seeds	0.0103	0.0090	0.0078	0.0171	0.0163	0.0156
7	Groundnuts	0.1946	0.1656	0.1366	0.3069	0.2827	0.2585
8	Kidney beans	0.5696	0.5143	0.4589	**1.7234**	**1.7121**	**1.7007**
9	Maize	**5.7491**	**5.2668**	**4.7846**	**19.4587**	**19.4409**	**19.4232**
10	Soybeans	0.0046	0.0031	0.0016	0.0068	0.0056	0.0044
11	White pepper	0.0131	0.0098	0.0066	0.0160	0.0133	0.0105

Values of hazard index > 1 are indicated in bold.

**Table 3 ijerph-18-05043-t003:** Carcinogenic hazard ratio with values greater than 1 from risk assessment of 9 pesticide residues in 11 food items collected in the 3 largest cities of Cameroon.

Residue	OSF	Food Item	Carcinogenic Hazard Ratio > 1			
			Average Exposure			95th Percentile Exposure
			Upper Bound	Medium Bound	Lower Bound	Upper Bound
Aldrin	17	Maize	12,312.94	12,312.94	12,312.94	38,883.96
		Black beans	1041.83	1039.51	1037.19	3653.47
		Kidney beans	830.28	830.28	830.28	3104.32
		Groundnuts	96.18	94.93	93.68	209.97
		Chili pepper	2.11	2.09	2.07	9.06
Dieldrin	16	Maize	16,038.25	15,944.25	15,850.25	58,395.59
		Black beans	242.15	171.52	100.90	363.23
		Kidney beans	215.41	116.54	17.66	227.77
		Groundnuts	26.70	25.35	24.01	83.68
		Chili pepper	1.20	1.15	1.10	6.68
HCB	1.6	Maize	1525.43	1363.57	1201.70	4453.17
		Black beans	32.69	32.69	32.69	71.74
		Kidney beans	18.82	18.82	18.82	31.10
		Groundnuts	8.32	4.62	0.93	9.72
		Chili pepper	0.31	0.31	0.31	1.21
Heptachlor	4.5	Maize	5041.98	4086.56	3131.15	15,137.05
		Black beans	92.75	81.63	70.52	328.18
		Kidney beans	45.55	32.21	18.87	90.45
		Groundnuts	11.35	5.67	0.00	11.35
		Chili pepper	0.89	0.75	0.61	2.61
o,p’-DDT	0.34	Maize	155.77	98.72	41.68	218.23
		Black beans	5.36	5.10	4.83	10.39
		Kidney beans	4.19	3.93	3.67	4.82
		Groundnuts	1.53	1.49	1.46	1.79
p,p’-DDD	0.24	Maize	116.73	116.73	116.73	205.11
		Black beans	3.21	2.08	0.95	4.89
		Kidney beans	2.97	2.08	1.18	3.68
p,p’-DDE	0.34	Maize	110.23	101.44	92.65	347.31
		Black beans	3.09	3.09	3.09	4.12
		Kidney beans	2.72	2.65	2.58	3.75
p,p’-DDT	0.34	Maize	532.30	477.21	422.12	990.81
		Black beans	33.55	33.42	33.29	69.81
		Kidney beans	26.36	25.96	25.56	57.50
		Groundnuts	5.62	5.54	5.46	8.25
β-HCH	1.8	Maize	961.89	563.23	164.57	1145.22
		Black beans	78.43	78.43	78.43	157.83
		Kidney beans	56.41	56.41	56.41	135.44
		Groundnuts	7.82	5.62	3.42	18.18
		Chili pepper	1.22	1.21	1.20	3.38

HCB = hexachlorobenzene, HCH = hexachlorocyclohexane, DEE = dichlorodiphenyldichloroethylene, DDD = dichlorodiphenyldichloroethane, DDT = dichlorodiphenyltrichloroethane. OSF = oral slope factor, from the integrated risk information system (IRIS) [56].

## Data Availability

Data supporting reported results can be found in the Appendix A.

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
