# Peer review of "Contamination of Foods from Cameroon with Residues of 20 Halogenated Pesticides, and Health Risk of Adult Human Dietary Exposure"

_ijerph, 2021, doi:10.3390/ijerph18095043_

Round 1
Reviewer 1 Report
The article "Contamination of foods from Cameroon with residues of 20 halogenated pesticides, and health risk of adult human dietary exposure" by Galani et al. demonstrated usages of various regulated and banned pesticides in Cameron. The authors also discussed similar patters of pesticides usages in other countries.
The paper can be published once the following comments are addressed.
One of the shortcomings of the paper is that, there is very little information about analytical experiment available for the readers. I would request the authors to include chromatographic conditions for GC-ECD including some spectra for the real sample analyses in this article.
Also, I think the article dedicated too many paragraphs discussing about other studies, that eventually takes away focus of the reader from the main subject of the paper i. e, pesticides usages in Cameron.
Discussion about other studies are welcome, but this should be concise, and its better to remember the fact that, this is not a review article. An example of excess can be found in lines 453-481.
Author Response
DETAILED RESPONSES TO REVIEWERS' COMMENTS
Dear reviewers, we are very thankful for your wonderful and thorough assessment of this manuscript, and for the valuable comments which greatly helped to improve the quality of this submission. All your comments and suggestions have been considered as explained in detail below. Edits are marked in red colour in the revised manuscript.
Reviewer 1
The article "Contamination of foods from Cameroon with residues of 20 halogenated pesticides, and health risk of adult human dietary exposure" by Galani et al. demonstrated usages of various regulated and banned pesticides in Cameron. The authors also discussed similar patters of pesticides usages in other countries.
The paper can be published once the following comments are addressed.
One of the shortcomings of the paper is that, there is very little information about analytical experiment available for the readers. I would request the authors to include chromatographic conditions for GC-ECD including some spectra for the real sample analyses in this article.
The chromatographic conditions for GC-ECD as well as residue extraction and clean up, and validation of the method have been added in section 2.2.
Examples of chromatograms of a standard mixture and a food sample have been added in section 3.1.1.
Also, I think the article dedicated too many paragraphs discussing about other studies, that eventually takes away focus of the reader from the main subject of the paper i. e, pesticides usages in Cameron.
Discussion about other studies are welcome, but this should be concise, and its better to remember the fact that, this is not a review article. An example of excess can be found in lines 453-481.
The discussion paragraphs have been shortened, keeping more focus just on Cameroon and closely related countries.
Reviewer 2 Report
- 38-48 In the light of the Stocholm convention, this statement is exagerated and should be refined. Perhaps refer to countries where these pesticides are still in use. It is unlikely that the use of any chemical restricted by the Stockholm convention increases in developed countries.
L.112 300 g sample cannot be considered representative and makes questionable the results
- 115 is it not a mistake +20°C???
- 125 Reference 30 provided validation data which did not meet the minimum criteria of acceptability. For instance: the claimed LOQ values were 10-100 time lower than the lowest fortification level (0.01). The correlation coefficient of calibration (linearity) was reported with 4 digits without giving the calibrated range. The LOQ values were not demonstrated with chromatograms. In the light of 270% recoveries very large matrix effect was present, which is quite normal for ECD detection (LC-MS/MS as well). Under this condition the reliability of reported results cannot be verified.
The LOQ values reported in the supplementary information are quite different from those presented in the article
- 134-136 SANCO GL (C43) states ‘A practical default range of 60-140 % may be used for individual recoveries in routine analysis. Recoveries outside the above mentioned range would normally require re-analysis of the batch, but the results may be acceptable in certain justified cases’. E.g. no residue is detected! However, in this case sometimes relative high residues were detected
Please note that it is not sufficient to validate the method once, its performance parameters should be verified regularly during the practical application of the method.
L 206-208 check the < and> signs, they are wrongly applied
L233 URL given for ref 52 does not work
L238. Three HI-s were calculated. Their difference cannot be tested with F-test. Describe what values were considered.
Section 312. The reference values (ADI, ARfD, MRLs) should be summarized in a table and placed in the revised main document
L634: supplementary information cannot be accessed at www.mdpi.com/1193204/s1

Author Response
DETAILED RESPONSES TO REVIEWERS' COMMENTS
Dear reviewers, we are very thankful for your wonderful and thorough assessment of this manuscript, and for the valuable comments which greatly helped to improve the quality of this submission. All your comments and suggestions have been considered as explained in detail below. Edits are marked in red colour in the revised manuscript.
Reviewer 2
38-48 In the light of the Stocholm convention, this statement is exagerated and should be refined. Perhaps refer to countries where these pesticides are still in use. It is unlikely that the use of any chemical restricted by the Stockholm convention increases in developed countries.
The sentence was refined to reflect that it is the use of many other OCPs that has been rising, i.e., not the ones banned in the Stockholm convention.
L.112 300 g sample cannot be considered representative and makes questionable the results
According to the FAO’s RECOMMENDED METHODS OF SAMPLING FOR THE DETERMINATION OF PESTICIDE RESIDUES FOR COMPLIANCE WITH MRLS CAC/GL 33, the amount of sample should be between 500g-1kg, depending on the commodity. But because of expensive cost to ship the samples to Belgium, we limited to approx. 300 g for each sample. However, each batch of commodity was well mixed before sampling.
115 is it not a mistake +20°C???
Not a mistake. The lab has a controlled room for keeping the food commodities at +20°C. Pesticide extracts instead were kept in the fridge at -20°C when awaiting analysis.
125 Reference 30 provided validation data which did not meet the minimum criteria of acceptability. For instance: the claimed LOQ values were 10-100 time lower than the lowest fortification level (0.01).
The LOQ was determined by the Two-step approach using t99sLLMV method, which includes calculations. In this case, LOQ can be lower than the spiked concentration. (Corley, J. Best practices in establishing detection and quantification limits for pesticide residues in foods. In Handbook of Residue Analytical Methods for Agrochemicals; Lee, P. W., Ed.; John Wiley & Sons Ltd.: Hoboken, NJ, 2003; Vol. 1-2, pp 59-75.)
The correlation coefficient of calibration (linearity) was reported with 4 digits without giving the calibrated range.
The calibration range (0.1, 0.05, 0.01, 0.005, 0.001 mg/L) has been added in section 2.2.4.
The LOQ values were not demonstrated with chromatograms.
As also requested by another reviewer, chromatograms have been added in section 3.1.1.
In the light of 270% recoveries very large matrix effect was present, which is quite normal for ECD detection (LC-MS/MS as well). Under this condition the reliability of reported results cannot be verified.
We do agree that high recovery values due to matrix effect are frequent in pesticide residue analysis. We explained in section 2.2.4 on which ground, and how this issue was handled.
The LOQ values reported in the supplementary information are quite different from those presented in the article
The LODs were between 0.0004 and 0.0652 mg/kg, and the LOQs varied from 0.0012 to 0.2180 mg/kg. These are in agreement with what is reported in Supplementary information 1, see column G for LOD and column H for LOQ.
134-136 SANCO GL (C43) states ‘A practical default range of 60-140 % may be used for individual recoveries in routine analysis. Recoveries outside the above mentioned range would normally require re-analysis of the batch, but the results may be acceptable in certain justified cases’. E.g. no residue is detected! However, in this case sometimes relative high residues were detected
Please note that it is not sufficient to validate the method once, its performance parameters should be verified regularly during the practical application of the method.
We completely agree with the reviewer; in the Laboratory of Crop Protection Chemistry of the Department of Plants and Crops, Faculty of Bioscience Engineering, Ghent University, performance equipment and analysis methods are routinely checked.
L 206-208 check the < and> signs, they are wrongly applied
Checked.
L233 URL given for ref 52 does not work
A new URL has been provided.
L238. Three HI-s were calculated. Their difference cannot be tested with F-test. Describe what values were considered.
HI lower bound vs HI medium bound
HI lower bound vs HI upper bound
HI medium bound vs HI upper bound
Section 312. The reference values (ADI, ARfD, MRLs) should be summarized in a table and placed in the revised main document
There are 23 residues and sum of residues, and 11 foods: that will be a table of 253 lines. This would be a huge table to add in the main document, presenting data which are not our results, which can be found in available databases on the internet, and which are already well summarized in the supplementary material, next to their counterpart for comparison. We think it is not useful, but if the reviewer an editor think otherwise, we are happy to generate that table and add it in the main document.
L634: supplementary information cannot be accessed at www.mdpi.com/1193204/s1
We have uploaded the files, and they are accessible to us. Please, request the Editor to provide you access.